# UES: An Ultra-expanded Space for Unsupervised Domain Adaptation

## Abstract

Unsupervised Domain Adaptation (UDA) offers a promising solution to address label annotation costs and dataset bias by facilitating knowledge transfer from a label-rich source domain to a related but unlabeled target domain. While the FC+Softmax+Cross Entropy loss has become the de facto standard for classification under the IID assumption, its performance degrades significantly under UDA's non-IID setting, where target domain features frequently violate decision boundaries, resulting in inter-class confusion. To overcome this limitation, we propose an innovative Distance Margin-based Ultra-Expanded Space (UES) loss, which encourages features to occupy an expanded representation space, thereby maintaining a safer distance from decision boundaries. Designed as a plug-and-play regularization term, UES can be seamlessly integrated into various classification-based UDA frameworks, offering exceptional simplicity by requiring only few lines of code and minimal hyperparameter tuning while reducing computational overhead. Extensive experiments demonstrate that our method achieves performance improvements in nearly all tested cross-domain tasks.

## 1 Introduction

Deep neural networks that rely on large-scale, well annotated training datasets have achieved significant success. In practical applications, sufficient training data is often time-consuming and costly, and due to domain transfer issues, the generalization ability to new datasets is poor (Pan & Yang, 2009). Therefore, Unsupervised Domain Adaptation (UDA) has emerged as a crucial research direction, focusing on leveraging knowledge from label rich source domains to facilitate learning of relevant but unlabeled target domains (Li et al., 2021b).

UDA has seen major progress in theory and algorithms, with most feature distribution alignment methods (Wang & Deng, 2018; Zhang et al., 2019;

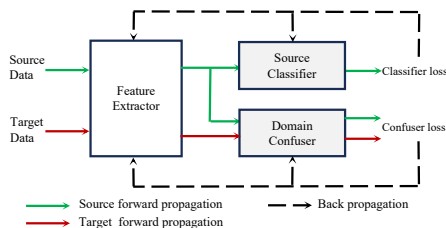

Figure 1: The generalized architecture for UDA. Different confusers represent different alignment algorithms.

Liu et al., 2022; Li et al., 2020) following a common three-part architecture in Figure 1. The core component is the feature extractor, which learns domain-invariant representations. The source classifier (the last layer is FC with softmax) handles supervised prediction for the source domain. The domain confuser module implements the alignment strategy to reduce domain differences. During training, the feature extractor processes source data for source classifier while feeding both source and target features into the domain confuser. Through backpropagation, this dual-path setup helps the model learn transferable representations for target domain.

We note two key issues.

The first challenge. Target domain features exhibit significantly greater intra-class variation compared to source domain features in the first column of the Figure 2, leading to decision boundary violations and class confusion due to higher epistemic uncertainty in OOD data (Vu et al., 2019). The conventional margin loss (FC+Softmax+Cross Entropy loss) module effectively maximizes source

domain log-likelihood in the probability space but fails to offer the generalization feature space required for target domain classification. Based on the foundation established in FC+Softmax+Cross Entropy loss, the enhanced margin loss (such as ArcFace (Deng et al., 2019) and Center loss (Wen et al., 2016) ) is proposed, which provides a solution by enhancing intra-class compactness and inter-class separability. However, these supervised and IID margin methods does not provide a broad space for target domain features in Figure 2 and promotes the occurrence of negative transfer phenomena (Jiang et al., 2023; Zhang et al., 2022). Although margin methods shows promising performance in IID, its potential have never blossomed in the UDA.

The second challenge. There is an asymmetric determination mechanism: source feature distribution emerge from the combined parameters influence of feature extractor, source classifier, and domain confuser, while target lack classifier parameters guidance (Li et al., 2024).

Our core contribution lies in enhancing feature discriminability through positioning features away from decision boundaries to prevent Inter-class confusion, while comprehensively exploiting the parametric representation capacity of classifiers. This work first derives a Distance Margin from the geometric properties of the FC+Softmax+Cross-Entropy framework, then formulates the Ultra-Expanded Space (UES) loss as a theoretically-grounded regularization term pushing the target domain error further below the generalization bound. The proposed UES loss is universally applicable to any classification paradigm employing linear classifiers—a design pattern ubiquitous across diverse domains, with typical applications including semantic segmentation (pixel-wise classification), object detection (region-based classification) and language modeling (token-level classification) (Azuma et al., 2023).

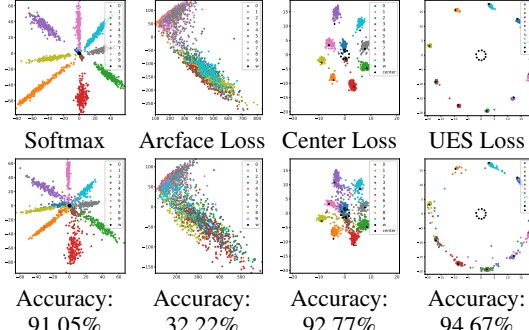

Figure 2: Feature visualization of source domain (MNIST, top) and target domain (USPS, bottom) under different margin losses and provide the accuracy of the target domain. Specifically, we appended a 2D bottleneck layer after the feature extractor and presented it by class. Softmax indicates only using UDA algorithm.

Our contributions are summarized as follows:

- We propose a margin-based UES loss, representing the first margin method specifically designed for Unsupervised Domain Adaptation (UDA). It constructs an ultra-expanded feature space that effectively pushes features away from decision boundaries, thereby preventing inter-class feature boundary violations.

- The proposed UES loss serves as a simple yet plug-and-play regularization term that can be seamlessly integrated into existing UDA classification frameworks. It requires minimal implementation effort (just few lines of code), demands simplified hyperparameter tuning, and introduces lower training overhead by enhancing computational graph utilization efficiency.

- Extensive experiments demonstrate that the UES regularization term achieves performance improvements across diverse cross-domain tasks, exhibiting remarkable generalization capability.

## 2 RELATED WORK

**UDA Distribution Alignment**. UDA distribution alignment methods can be categorized into three primary strategies based on their operational spaces: Input space, the performance degradation on the target domain primarily stems from the distribution shift between the source and target data. Cross-domain data augmentation addresses this issue by synthesizing target-style representations from source samples(Hoffman et al., 2018; Li et al., 2021a; Chen et al., 2025; Dong et al., 2025); Output space, developed from self-supervised learning and self-training to provide additional supervisory signals (Mei et al., 2020; Liu et al., 2021; Li et al., 2022; Yuan et al., 2024; Ali et al., 2025); and Feature space-currently the most prevalent due to their simplicity and efficiency-focusing on

feature distribution alignment (Wu et al., 2021; Wei et al., 2021; Dan et al., 2023; Zhou et al., 2023; Fang et al., 2024; Na et al., 2025), with the first two approaches can be combined with this method (Shen et al., 2023).

**Generalization Error Bound**. Theoretical frameworks that constrain target domain error within generalization bounds have driven significant algorithmic advances. Ben-David et al. (2006) established the first domain adaptation bound, which Ben-David et al. (2010) extended through analysis in a symmetric hypothesis space. Zhao et al. (2019) demonstrated that algorithms learning domain-invariant features based on such theories may fail in certain scenarios, and proposed an alternative bound based on label function discrepancy. Addressing the gap between binary theory and multi-class practice, Zhang et al. (2019) developed a margin loss-based bound. In source-free domain adaptation, Gezheng et al. (2024) applied contrastive learning to establish a unified upper bound for sample discrepancy metrics on outlier-robust risk. Shen et al. (2023) introduced a bias-variance trade-off approach to control generalization bounds in multi-source-free domain adaptation.

**Margin Loss**. In classification problems, margin refers to a function that characterizes the confidence disparity of predictions between different classes, and margin loss denotes an optimizable objective function that governs this margin. Among various approaches, the combination of FC+Softmax and Cross Entropy loss has emerged as the most widely adopted and effective paradigm. The former components quantify inter-class probability differences, whereas the latter optimizes these differences. To further enhance intra-class compactness and inter-class separability within this FC+Softmax+Cross-Entropy loss, two major methodological directions have been developed. First, the weight matrices of the FC layer and features are normalized and inner-producted to obtain the angular cosine, which is made directionally more discriminative by adjusting the angular margin of the different categories (Liu et al., 2017; Wang et al., 2018b;a; Kim et al., 2022; Xu et al., 2024). Second, a center is learned for each category, and all the features of each category are pulled towards the corresponding category center based on the Euclidean metric (Wan et al., 2018; Pang et al., 2019; Zhang et al., 2020; Yang et al., 2022; Peng et al., 2025). These methods fail to provide features with a more expansive representation space, consequently limiting their effectiveness in enhancing UDA performance.

## 3 PRELIMINARIES

This section will introduce the basic notations in UDA classification problems and a generalization bound.

**Symbol**. In UDA, there are two domains accessible: a labeled source domain containing $N_s$ samples, denoted as $\mathcal{S} = \{(x_i^s, y_i^s)\}_{i=1}^{N_s}$, where $y_i^s \in \{1, 2, ..., K\}$ represents the label of the $i$-th source sample $x_i^s$; and an unlabeled target domain comprising $N_t$ samples, represented as $\mathcal{T} = \{(x_i^t)\}_{i=1}^{N_t}$. While both domains share the same label space, they exhibit divergence in their underlying probability distributions. This distributional discrepancy typically leads to performance degradation when a model trained solely on the source domain is directly applied to the target domain.

**Margin generalization boundary**. Zhang et al. (2019) presents a generalization bound based on the margin loss, extending previous bounds that were confined to binary classification to practical multi-class problems. The expected error $err_{\mathcal{T}}(f)$ on the target domain is bounded by the sum of four terms: the empirical error $err_{\hat{\mathcal{S}}}^{(\rho)}(f)$ on the source domain, the empirical margin disparity discrepancy (MDD) $d_{f,\mathcal{F}}^{(\rho)}\left(\hat{\mathcal{S}}, \hat{\mathcal{T}}\right)$, an ideal empirical error term $\lambda^*$ and a complexity term. It is assumed that with a suitable architecture or design, the latter two terms can be made sufficiently small.

$$err_{\mathcal{T}}(f) \leq err_{\hat{\mathcal{S}}}^{(\rho)}(f) + d_{f,\mathcal{F}}^{(\rho)}\left(\hat{\mathcal{S}}, \hat{\mathcal{T}}\right) + \lambda^* + complexity \tag{1}$$

Here, the hypothesis space $\mathcal{F}$ consists of classifiers $f : x \to \mathbb{R}^{|y|} = \mathbb{R}^k$, where each dimension of the output represents the predictive confidence for a class. The superscript $(\rho)$ denotes a custom margin loss. For instance, $err_{\hat{\mathcal{S}}}^{(\rho)}(f)$ employs the Softmax+Cross-Entropy loss as the margin loss.

The problem then reduces to minimizing $d_{f,\mathcal{F}}^{(\rho)}\left(\hat{\mathcal{S}}, \hat{\mathcal{T}}\right)$, formulated as follows:

$$d_{f,\mathcal{F}}^{(\rho)}\left(\hat{\mathcal{S}}, \hat{\mathcal{T}}\right) \triangleq \sup_{f' \in \mathcal{F}}\left(disp_{\hat{\mathcal{T}}}^{(\rho)}\left(f', f\right) - disp_{\hat{\mathcal{S}}}^{(\rho)}\left(f', f\right)\right) \tag{2}$$

$$disp_{\hat{\mathcal{D}}}^{(\rho)}\left(f', f\right) \triangleq \mathbb{E}_{\hat{D}} MarginLoss_{f'}\left(, h_f\right) = \frac{1}{n}\sum_{i}^{n} MarginLoss_{f'}\left(x_i, h_f\left(x_i\right)\right) \tag{3}$$

Here, $h_f$ denotes the class predicted with the highest probability by $f$, where $f$ is the source domain classifier and $f'$ is an auxiliary target domain classifier. By introducing a feature extractor $\varphi$, the minimization of MDD is formulated as a minimax game:

$$\min_{f,\varphi} \max_{f'}\left(disp_{\varphi(\hat{\mathcal{T}})}^{(\rho)}\left(f', f\right) - disp_{\varphi(\hat{\mathcal{S}})}^{(\rho)}\left(f', f\right)\right) \tag{4}$$

## 4 METHOD DERIVATION AND ANALYSIS

This chapter is structured as follows. We commence by introducing a distance margin within the feature space in Section 4.1. Building upon this, Section 4.2 integrates the proposed distance margin with the generalization bound from Section 3.2, thereby formulating the UES loss. This UES loss is a margin-based regularization function designed to be universally applicable to UDA methods involving classification. Section 4.3 furnishes a theoretical demonstration that the UES loss surpassesthe commonly employed entropy minimization in UDA. The overall formulation of the proposed approach is consolidated in Section 4.4.

### 4.1 DISTANCE MARGIN

To enhance the discriminability of sample features, it is imperative to position them as far as possible from the decision boundary. As illustrated in Figure 3, when analyzing with two adjacent templates $W_1$ and $W_2$ (where a template $W_k$ denotes the class vector corresponding to label $k$ in the final linear classification layer $W$ ), the decision boundary $v$ between the two classes must satisfy $W_1^T v = W_2^T v$ (ignoring the bias term), which is equivalent to $(W_1^T - W_2^T)v = 0$. The distance from a sample feature $a_i$ to this decision boundary is given by:

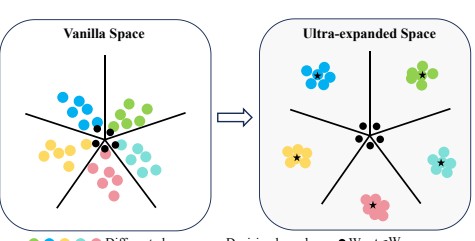

Figure 3: (Best viewed in color.) Schematic diagram of the Distance Margin mechanism.

$$Distance = \frac{\left\|\left(W_1^T - W_2^T\right) \cdot a_i\right\|_2}{\left\|\left(W_1^T - W_2^T\right)\right\|_2} \tag{5}$$

To maximize this distance, an intuitive and elegant approach is to scale the sample feature $a_i$ by a factor of $\rho$, yielding:

$$Distance_\rho = \frac{\left\|\left(W_1^T - W_2^T\right)\left(\rho a_i\right)\right\|_2}{\left\|\left(W_1^T - W_2^T\right)\right\|_2} = \rho\frac{\left\|\left(W_1^T - W_2^T\right) a_i\right\|_2}{\left\|\left(W_1^T - W_2^T\right)\right\|_2} = \rho Distance \tag{6}$$

To achieve this effect, we employ the classifier's margin loss, which integrates a fully-connected layer, softmax activation, and cross-entropy loss (Appendix A):

$$softmax\left(Wa_i + b\right)_{y_i} = \frac{e^{W_{y_i}^T a_i + b_{y_i}}}{\sum_k e^{W_k^T a_i + b_k}} = \frac{e^{-\left\|a_i - W_{y_i}\right\|_2^2 + \left\|W_{y_i}\right\|_2^2 - W_{y_i}^T a_i + b_{y_i} + \left\|a_i\right\|_2^2}}{\sum_k e^{-\left\|a_i - W_k\right\|_2^2 + \left\|W_k\right\|_2^2 - W_k^T a_i + b_k + \left\|a_i\right\|_2^2}} \tag{7}$$

We observe that while minimizing the margin loss reduces the Euclidean distance between the feature and its corresponding template, the complex functional relationship and variable dependencies between the margin loss and the distance $\left\|a_i - W_{y_i}\right\|_2^2$ prevent it from effectively enhancing intra-class compactness. Consequently, we adopt the distance $\left\|a_i - W_{y_i}\right\|_2^2$ directly as a feature-level

supervisory signal. However, this alone does not strengthen inter-class separability. The afore-mentioned decision boundary analysis provides a solution. We thus propose a Distance Margin as follows:

$$Margin_{Distance} = \|a_i - \rho W_{y_i}\|_2^2 = \|\varphi(x_i) - \rho W_{y_i}\|_2^2 \tag{8}$$

where $\rho$ is a Ultra-expanded hyperparameter, and the gradient computation for $W_{y_i}$ is detached. Under this formulation, When $\rho$ assumes large values while $Margin_{Distance}$ undergoes reduction, the transformed feature $\varphi(x_i)$ is compelled to approach $\rho W_{y_i}$, thereby constructing a Ultra-expanded space that promotes both intra-class compactness and inter-class separability.

## 4.2 ULTRA-EXPANDED SPACE LOSS

Substituting the distance margin from Equation 8 into Equation 3 yields:

$$disp_{\hat{D}}^{(\rho)}(f', f) \triangleq \frac{1}{n} \sum_{i=1}^{n} \left\|\varphi(x_i) - \rho W_{h_f(x_i)}^{f'}\right\|_2^2 \tag{9}$$

Where $W^{f'}$ denotes the class weight matrix of the auxiliary target domain classifier $f'$. Since we detach the gradient computation for $W$, this constitutes a function independent of the optimization of classifiers $(f', f)$. Consequently, Equation 4 can be reformulated as follows:

$$\min_{f,\varphi} \max_{f'} \left( disp_{\varphi(\hat{\mathcal{T}})}^{(\rho)}(f', f) - disp_{\varphi(\hat{\mathcal{S}})}^{(\rho)}(f', f) \right)$$

$$= \min_{f,\varphi} \max_{f'} \left( \frac{1}{N_t} \sum_{i=1}^{N_t} \left\|\varphi(x_i^t) - \rho W_{h_f(x_i^t)}^{f'}\right\|_2^2 - \gamma \frac{1}{N_s} \sum_{i=1}^{N_s} \left\|\varphi(x_i^s) - \rho W_{h_f(x_i^s)}^{f'}\right\|_2^2 \right) \tag{10}$$

$$= \min_{\varphi} \left( \frac{1}{N_t} \sum_{i=1}^{N_t} \left\|\varphi(x_i^t) - \rho W_{h_f(x_i^t)}^{f'}\right\|_2^2 - \gamma \frac{1}{N_s} \sum_{i=1}^{N_s} \left\|\varphi(x_i^s) - \rho W_{h_f(x_i^s)}^{f'}\right\|_2^2 \right)$$

where $\gamma$ is a coefficient that combines two terms. As $W^{f'}$ not undergo gradient-based optimization, we directly employ the parameters $W^f$ from the source domain classifier. This implementation effectively reduces the framework to a single-classifier architecture. In practice, we observe that the latter term yields fluctuating performance gains. This is likely because the source domain features are already well-clustered by the supervised loss of the source classifier, rendering this additional regularization term redundant or even conflicting. Thus, we set $\gamma = 0$. The optimization objective is thus formulated as the minimization of the UES loss.

$$\min UES = \min_{\varphi} \left( \frac{1}{N_t} \sum_{i=1}^{N_t} \left\|\varphi(x_i^t) - \rho W_{h_f(x_i^t)}^{f}\right\|_2^2 \right) \tag{11}$$

Divested of its minimax game mechanism, the UES loss is functionally redefined as a model-agnostic regularizer, enabling direct incorporation into arbitrary UDA architectures. This adaptability makes it applicable to a wide range of UDA classification tasks, such as image classification, semantic segmentation and object detection.

Conventional UDA methods aim to bound the target domain error via a generalization bound and then minimize this bound. In contrast, our proposed margin-based regularization not only operates within this theoretical framework but also further pushes the target domain error away from the established generalization bound, thereby achieving additional performance gains.

## 4.3 DISTANCE MARGIN V.S. ENTROPY MINIMIZATION

We proceed with a formal analysis of the properties of the distance margin in Equation 8. When optimizing the standard FC + softmax + cross-entropy loss, the feature vector $a_i = \varphi(x_i)$ tends to distribute around a scaled version of the template vector $\rho_w W_{y_i}$, where the scaling factor $\rho_w$ is not fixed (Figure 2 helps to understand qualitatively). By introducing and effectively optimizing the proposed loss from Equation 8, the feature $a_i$ is constrained to a region $\rho W_{y_i}$ near, which is equivalent to scaling $a_i$ by a factor of $\rho$. Therefore, Equation 7 becomes $softmax\left(W\frac{\rho}{\rho_w}a_i + b\right)$ to

produce a clearer output probability distribution. It is noteworthy that entropy minimization $H(p) = -plogp$, also promotes sharper probability distributions. The gradients of these two objectives with respect to the feature $a_i$ are given by:

$$2(a_i - \rho W_{y_i})$$
$$W(p(H(p) - logp)) \tag{12}$$

Obviously, the former has a more stable gradient that makes learning easier and prevents overfitting.

Entropy minimization is widely adopted in UDA, where it is generally understood to enhance the confidence of predictions on the target domain or to encourage more similar prediction distributions across domains, thereby promoting domain adaptation. Based on the theoretical analysis above, the proposed UES regularization—rooted in the distance margin—represents a theoretically superior alternative to conventional entropy minimization.

### 4.4 OVERALL FORMULATION

The overall objective function is formulated as follows:

$$\mathcal{L} = \mathcal{L}_{UDA} + \lambda \mathcal{L}_{UES} \tag{13}$$

The proposed margin regularization method can be directly incorporated into existing UDA frameworks. The complete implementation Pytorch code for $\mathcal{L}_{UES}$ is provided in Figure 4(right). This work introduces only two hyperparameters: a standard trade-off parameter $\lambda$ and the Ultra-expanded hyperparameter $\rho$ inherent to the $\mathcal{L}_{UES}$ formulation.

## 5 EXPERIMENTS

### 5.1 EXPERIMENTAL SETTING

**Dataset**. The evaluation employs four different difficulty benchmarks for image classification tasks: (1) Digit [USPS (1.8k)/MNIST (2k images), 10 classes(M/U), 2 tasks]; (2) Office-31 [4.1k images, 31 classes (A/W/D), 6 tasks]; (3) Office-Home [15.5k images, 65 classes (Ar/Cl/Pr/Rw), 12 tasks]; and(4) DomainNet[600K images, 345classes (C/I/P/Q/R/S), 30 tasks].

Semantic segmentation task is evaluated via adaptation from the GTA5 dataset (24,966 images) to the CityScapes dataset (2,975 training and 500 validation images) across the 19 shared categories.

**UDA Algorithm**. We evaluate five UDA algorithms for UDA: DAN(Long et al., 2015), Deep-CORAL(Sun & Saenko, 2016), DANN(Ganin & Lempitsky, 2015), DAAN(Yu et al., 2019) and DSAN(Zhu et al., 2020).

**Margin loss**. In addition to the proposed UES loss, the present study employs Center loss and ArcFace loss as baseline comparative methods, with all hyperparameters strictly following the configurations established in their respective original publications. This approach ensures a fair and reproducible experimental setup while maintaining methodological consistency with existing literature.

**Training Details**. For fair comparison with existing methods, we employed the standard ResNet-50 pretrained on ImageNet as the backbone network. Experiments were conducted with three different random seeds (0, 1, 2) and results were averaged. For Office-31 and Office-Home datasets, input images were resized to $224 \times 224$, while Digit dataset images were resized to $16 \times 16$. Notably, for the Digit dataset, we modified the ResNet-50 architecture by replacing its first convolutional and pooling layers with a $1 \times 1$ convolutional layer. Following Wang et al. (2025)'s optimization strategy, we used SGD with momentum 0.9, weight decay $5 \times 10^{-4}$ , and an initial learning rate of 0.1 which was decayed according to $(1 + 0.0003 * epoch)^{-0.75}$.

### 5.2 RESULTS

**UD Single-Source Image Classification**. Experimental results of Digit,Office31 and OfficeHome are presented in appendix B and a more concise performance change chart is drawn in Figure 4 (a) (The x-axis data represents the baseline performance using only UDA algorithms, while the y-axis

```python
# Pass properly-sized feature, W, logits into UESLOSS
import torch.nn as nn
def UESLOSS(self,feature_t,W,logits_t,lambd:float):# feature:[N,D]  W:[K,D]  logits:[N,K]
    ues_loss=nn.MSELoss()(feature_t,(0.132*W.shape[1]+123)*W[logits_t.argmax(dim=1)].detach())
    # D≥2048 for better performance; lambd: task-specific hyperparameter
    return lambd*ues_loss
```

Figure 4: (left)Performance changes under different margin loss. (right) Plug and play Pytorch UES code.

| UDA | Margin | R→C | R→P | P→C | P→R | C→S | S→P | Avg |
|---|---|---|---|---|---|---|---|---|
| DANN | Softmax | 49.13 | 47.05 | 45.41 | **59.75** | 41.07 | 42.93 | 47.56 |
| | UES | **54.09** | **49.24** | **46.39** | 59.41 | **42.50** | **44.54** | **49.36** |
| DSAN | Softmax | 52.69 | 49.61 | 45.74 | 58.90 | 42.74 | 44.53 | 49.04 |
| | UES | **55.91** | **51.19** | **46.97** | **59.50** | **43.82** | **45.27** | **50.43** |

Table 1: Accuracy (%) on DomainNet for unsupervised domain adaptation

represents the performance after using margin regularization loss. The diagonal line serves as a reference, where points above the line indicate performance improvement and points below denote degradation. Data points are color-coded by their respective margin loss for clear visualization). The results demonstrate that: (1) Arcface loss only achieves accuracy improvement on the simpler Digit dataset; (2) Center loss exhibits accuracy fluctuations around the baseline performance across all three datasets; and (3) our proposed UES loss consistently improves accuracy across all datasets and feature alignment methods, significantly outperforming both Arcface and Center loss. A representative example is the A→W task in Table 5, where our UES loss achieves over 10% accuracy improvement while both ArcFace and Center loss show decreased performance.

The partial experimental results of the DomainNet dataset are shown in Table 1, which is a dataset with significant differences between domains. We found an example of a slight decrease in accuracy when using UES. UES leverages pseudo-labels generated to push target-domain features away from source decision boundaries. However, its effectiveness can be compromised when pseudo-labels are of low quality. The pseudo-label weighting technique, commonly employed in self-training, can be integrated into the UES framework and is expected to yield further performance gains.

**UDA Multi-Source Image Classification**. We evaluated the OfficeHome dataset using the multi-

| | Cl,Pr,Rw→Ar | Ar,Pr,Rw→Cl | Ar,Cl,Rw→Pr | Ar,Cl,Pr→Rw | Avg |
|---|---|---|---|---|---|
| MFSAN | 71.24 | 60.8 | 80.58 | 81.66 | 73.57 |
| MFSAN+UES | **72.68** | **62.1** | **81.48** | **82.71** | **74.74** |

Table 2: Accuracy (%) on OfficeHome for multi-source unsupervised domain adaptation

source unsupervised domain adaptation algorithm MFSAN (Zhu et al., 2019), directly applying the hyperparameters from Figure 4(right) without any modifications. The results in Table 2 demonstrate that UES loss consistently enhances performance across every experimental configuration.

**UDA Semantic Segmentation**. We evaluate the GTA5→CityScapes adaptation using the Advent algorithm with DeepLabV3 as the baseline. In DeepLabV3, the output is generated through a final convolutional layer followed by interpolation-based upsampling. This convolutional layer is treated as a linear layer for applying the UES loss, and we adjust its hidden dimension to 2048. All trade-off hyperparameters of Advent were set to 0.001, and the same trade-off parameter was applied to the UES without any modifications. The results are illustrated in Figure 5.

## 5.3 ANALYSIS

**Convergence and Runtime**. As illustrated in Figure 6(left), the well-behaved landscape of the loss function, due to its convexity and smoothness, results in a smoother trajectory of accuracy convergence, making the learning process more robust.

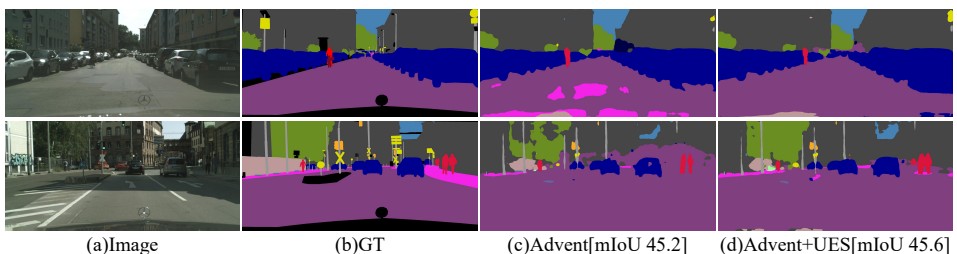

|        (a)Image        |        (b)GT        |   (c)Advent[mIoU 45.2]   |   (d)Advent+UES[mIoU 45.6]   |

Figure 5: Results in the GTA5→Cityscapes .

A counterintuitive observation is that although the UES loss introduces an additional tensor operation—which would theoretically lead to a slight increase in computational time—the empirical results indicate an actual reduction in runtime. This improvement can be attributed to the fact that the UES loss may enable the framework's Pytorch computational graph optimizer to identify and exploit more opportunities for operator fusion, thereby enhancing overall execution efficiency.

**Hyperparameter**. As shown in Figure 6, we conduct sensitivity analysis on the two newly introduced hyperparameters, $\lambda$ and $\rho$. $\lambda$ serves as a conventional balancing coefficient and demonstrates stable performance when values remain below 1. We recommend adjusting this parameter according to different task types—for instance, setting $\lambda = 0.1$ has proven effective for image classification tasks. More importantly, the Ultra-expanded parameter $\rho$ is tested across an extensive range from 10 to 3000, where it consistently yields stable and satisfactory results.

To further investigate its scalability across different feature dimensions, we examine the relationship between the optimal $\rho$ and the dimensionality of $W$ (Figure 6, right). While previous experiments were conducted using 2048-dimensional features from ResNet, we observe that as the dimensionality increases, the optimal $\rho$ value exhibits an approximately linear growth trend. By taking the average of the top three $\rho$ values corresponding to the highest accuracy at each dimensionality and applying least squares fitting, we derive the relationship $\rho = 0.132\dim(W) + 123$. This establishes $\rho$ as an automatically configurable parameter, effectively eliminating the need for manual tuning in practical applications. Based on observations, we find that smaller dimensions of the $W$ tend to lead to significant performance fluctuations across different values of the hyperparameter $\rho$. Therefore, we recommend that the dimensionality of the linear projection $W$ should not be set too small in practice.

**UES v.s. Entropy Minimization**. We evaluate the performance of the DANN algorithm combined with UES and Entropy Minimization respectively across three distinct tasks in Table 3. The results demonstrate that UES consistently achieves superior performance.

|                     | Digit M→U | Office31 A→W | OfficeHome Rw→Pr |
|---------------------|-----------|--------------|------------------|
| Entropy Minimization | 94.11     | 89.55        | 81.48            |
| UES                 | **96.64** | **92.24**    | **82.61**        |

Table 3: Comparison of accuracy between UES and Entropy Minimization in three tasks

## 6 EFFECTIVENESS ANALYSIS OF THE UES

To elucidate the underlying mechanisms for the effectiveness of UES, we proceed with a systematic analysis from two perspectives: the optimization behavior of the loss, and the enhancement of intra-class compactness alongside inter-class separability.

**Optimization behavior of the loss**. As the domain confuser becomes increasingly optimized, the influence of its loss signal on the encoder gradually weakens (Zhang et al., 2023),thereby impeding effective domain adaptation. As shown in Figure 6(middle), the proposed UES loss not only introduces auxiliary supervision to facilitate model optimization but also further reduces the domain confuser loss. A comparative analysis with two alternative margin-based approaches—Center loss and ArcFace loss—reveals distinct behavioral differences: Center loss exhibits a negligible reg-

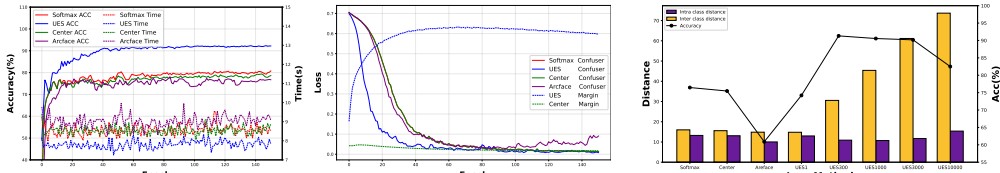

Figure 6: Schematic illustration of Office31 A→W task performance evolution across training epochs (left)Accuracy and runtime. (middle) Domain confuser and margin regularization term loss. ArcFace constitutes a direct variant of the Softmax, wherein neither formulation incorporates an explicit margin regularization term. (right) Intra-class and Inter-class distance in epoch 40. The notation UES$\rho$ denotes the method utilizing different Ultra-expanded hyperparameters$\rho$.

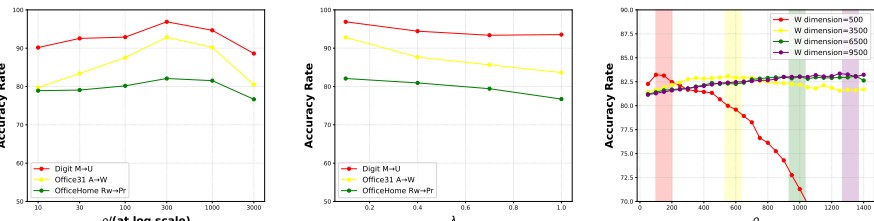

Figure 7: Analysis of hyperparameter sensitivity. (left)Fixed $\lambda = 0.1$. (middle)Fixed $\rho = 300$. (right)Performance Variation with Respect to $\rho$ and dim($W$).

ularization effect, manifested by virtually identical domain confuser loss curves compared to the baseline Softmax (evidenced by the overlapping green and red solid lines); ArcFace, as a Softmax variant, induces oscillatory and persistently increasing confuser loss.(Liang et al., 2021)

**Intra-class compactness and Inter-class separability** Intra-class compactness reflects the degree of aggregation of features within each class, and Inter-class separability quantifies the differences between different classes. We conduct qualitative and quantitative analyses on intra-class compactness and inter-class separability.

Qualitative Analysis: As shown in Figure 8, T-SNE is used to visualize the feature distribution. The proposed UES successfully forms 31 well-separated clusters with clear decision boundaries, leaving almost no scattered data points in the gaps between clusters.

Quantitative Analysis: Currently, there is no established method to quantify Intra-class compactness and Inter-class separability. We propose a novel approach based on their inherent

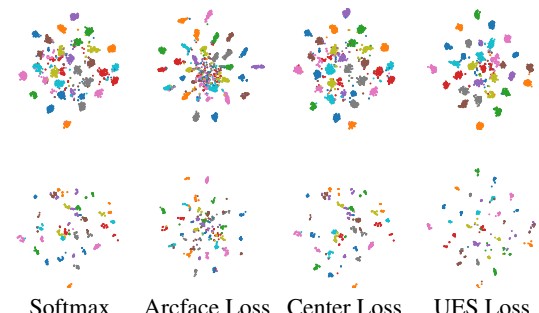

| Softmax | Arcface Loss | Center Loss | UES Loss |

Figure 8: In the transfer task Office 31 A→W, T-SNE visualization of learning features is performed, with the first row representing the source domain and the second row representing the target domain, with each column representing a different loss.

properties: Inter-class separability is measured by the average distance between class centers (termed inter-class distance), and intra-class compactness is evaluated by the average distance from features to their corresponding class centers based on ground-truth labels (termed intra-class distance). As illustrated in Figure 6 (right), the intra-class distance exhibits no significant correlation with the final accuracy, whereas the inter-class distance within a certain range can substantially enhance the accuracy. Notably, a larger Intra-class distance is not always beneficial; an excessively large intra-class distance leads to performance degradation, as it disrupts the semantic information among features. Both Center loss and ArcFace loss have minimal impact on inter-class and Intra-class distances,

while our proposed method significantly increases the Inter-class distance with flexible controllability.

The remarkable achievement in Inter-class separability highlights the extraordinary capability of UES in constructing an Ultra-expanded Space for feature.

## 7 CONCLUSION

This paper presents the first distance margin-based regularization loss specifically designed to address the issue of target domain data violating decision boundaries in Unsupervised Domain Adaptation (UDA). In contrast to prevailing margin-based methods, our approach achieves performance improvements by substantially enhancing Inter-class separability. The proposed regularization technique is universally applicable to any classification-based UDA framework and requires minimal hyperparameter tuning.

### ETHICS STATEMENT

The text of this manuscript was polished for English language using Deepseek. The authors are solely responsible for the final content.

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

# A CLASS WEIGHT MATRIX

The inner product operation between sample features and the class weight matrix (within the fully-connected layer), followed by projection into the probability space via the softmax function, enables the computation of cross-entropy loss with respect to the ground-truth labels. The class weight matrix, learned through the training of the entire neural network on the dataset, can be regarded as a high-order statistical representation of the data distribution across different classes. However, this information remains underutilized in the target domain. A natural intuition therefore suggests clustering features in the vicinity of their predicted class weight vectors.

# B EXPERIMENTAL RESULTS

| UDA | Margin | M→U | U→M | Avg |
|---|---|---|---|---|
| DAN | Softmax | 93.98 | 83.70 | 88.84 |
| | Arcface | 94.83 | 90.46 | 92.65 |
| | Center | 94.12 | 85.35 | 89.73 |
| | UES | **96.22** | **93.81** | **95.01** |
| DANN | Softmax | 92.29 | 87.53 | 89.91 |
| | Arcface | 95.07 | 90.38 | 92.72 |
| | Center | 91.74 | 86.78 | 89.26 |
| | UES | **96.64** | **93.18** | **94.91** |
| DeepCORAL | Softmax | 91.68 | 80.71 | 86.20 |
| | Arcface | 95.72 | 89.08 | 92.40 |
| | Center | 90.09 | 76.11 | 83.10 |
| | UES | **97.16** | **95.63** | **96.40** |
| DAAN | Softmax | 89.87 | 79.56 | 84.71 |
| | Arcface | 95.75 | 90.73 | 93.24 |
| | Center | 89.53 | 75.28 | 82.41 |
| | UES | **97.27** | **95.15** | **96.21** |
| DSAN | Softmax | 92.12 | 87.35 | 89.73 |
| | Arcface | 94.20 | 86.08 | 90.14 |
| | Center | 93.01 | 89.63 | 91.32 |
| | UES | **97.68** | **95.31** | **96.49** |

Table 4: Accuracy (%) on Digit for unsupervised domain adaptation

| UDA | Margin | A→W | W→A | A→D | D→A | D→W | W→D | Avg |
|---|---|---|---|---|---|---|---|---|
| DAN | Softmax | 82.72 | 65.74 | 82.32 | 67.41 | 97.82 | 100.00 | 82.67 |
| | Arcface | 78.49 | 61.68 | 76.77 | 63.94 | 97.90 | 99.93 | 79.78 |
| | Center | 81.55 | 65.47 | 81.32 | 66.73 | 97.65 | 99.93 | 82.11 |
| | UES | **92.20** | **70.87** | **90.36** | **71.37** | **98.44** | **100.00** | **87.21** |
| DANN | Softmax | 80.33 | 65.80 | 81.92 | 66.53 | 98.11 | 100.00 | 82.11 |
| | Arcface | 72.45 | 61.59 | 73.76 | 62.45 | 97.69 | 100.00 | 77.99 |
| | Center | 78.99 | 65.47 | 80.78 | 66.14 | 98.02 | 100.00 | 81.57 |
| | UES | **92.24** | **69.47** | **89.69** | **71.54** | **98.28** | **100.00** | **86.87** |
| DeepCORAL | Softmax | 79.95 | 66.05 | 81.79 | 66.89 | 97.98 | 100.00 | 82.11 |
| | Arcface | 71.78 | 61.88 | 73.42 | 62.90 | 97.90 | 99.86 | 77.96 |
| | Center | 79.11 | 65.74 | 80.92 | 66.47 | 97.73 | 99.82 | 81.63 |
| | UES | **91.82** | **69.12** | **89.35** | **70.95** | **98.61** | **100.00** | **86.64** |
| DAAN | Softmax | 76.77 | 65.53 | 81.05 | 66.25 | 97.82 | 100.00 | 81.23 |
| | Arcface | 71.02 | 60.91 | 73.02 | 62.32 | 97.94 | 100.00 | 77.53 |
| | Center | 75.51 | 65.11 | 80.92 | 65.68 | 97.48 | 100.00 | 80.78 |
| | UES | **91.48** | **68.76** | **88.55** | **70.37** | **98.53** | **100.00** | **86.28** |
| DSAN | Softmax | 84.90 | 66.86 | 82.86 | 69.21 | 98.53 | 99.93 | 83.71 |
| | Arcface | 69.89 | 61.25 | 73.02 | 63.34 | 98.11 | 99.86 | 77.58 |
| | Center | 77.02 | 66.11 | 78.11 | 68.59 | 98.36 | 99.86 | 81.34 |
| | UES | **91.94** | **69.38** | **92.50** | **70.80** | **98.69** | **100.00** | **87.22** |

Table 5: Accuracy (%) on office-31 for unsupervised domain adaptation

| UDA | Margin | Ar→Cl | Cl→Ar | Ar→Pr | Pr→Ar | Pr→Cl | Cl→Pr | |
|---|---|---|---|---|---|---|---|---|
| DAN | Softmax | 52.80 | 56.58 | 69.29 | 56.62 | 48.24 | 65.86 | |
| | Arcface | 44.18 | 48.96 | 57.53 | 47.61 | 43.85 | 55.08 | |
| | Center | 52.76 | 56.11 | 68.76 | 56.61 | 48.36 | 64.78 | |
| | UES | **56.15** | **62.94** | **71.81** | **62.69** | **54.18** | **70.47** | |
| DANN | Softmax | 51.88 | 55.25 | 68.44 | 55.29 | 47.80 | 63.99 | |
| | Arcface | 44.30 | 48.83 | 56.32 | 48.05 | 42.80 | 54.96 | |
| | Center | 52.05 | 55.21 | 68.19 | 54.89 | 47.58 | 64.51 | |
| | UES | **55.47** | **60.07** | **71.29** | **60.05** | **53.67** | **69.42** | |
| DeepCORAL | Softmax | 51.89 | 55.72 | 69.58 | 55.89 | 46.78 | 65.72 | |
| | Arcface | 44.29 | 48.35 | 56.39 | 46.91 | 42.60 | 55.25 | |
| | Center | 51.96 | 55.54 | 68.99 | 55.50 | 46.52 | 65.39 | |
| | UES | **54.76** | **62.38** | **71.81** | **61.87** | **52.99** | **70.48** | |
| DAAN | Softmax | 51.24 | 55.12 | 68.39 | 54.52 | 45.90 | 64.87 | |
| | Arcface | 44.36 | 48.96 | 56.65 | 46.79 | 42.27 | 55.25 | |
| | Center | 51.34 | 54.99 | 68.32 | 55.23 | 45.68 | 64.66 | |
| | UES | **54.25** | **61.92** | **71.53** | **61.50** | **53.40** | **70.04** | |
| DSAN | Softmax | 53.69 | 57.02 | 69.72 | 60.11 | 53.76 | 67.40 | |
| | Arcface | 48.75 | 54.34 | 63.03 | 52.57 | 47.39 | 59.22 | |
| | Center | 53.90 | 56.90 | 69.43 | 60.44 | 54.04 | 67.31 | |
| | UES | **56.88** | **67.71** | **71.27** | **61.68** | **55.32** | **70.55** | |

| Alignment | Margin | Ar→Rw | Rw→Ar | Cl→Rw | Rw→Cl | Pr→Rw | Rw→Pr | Avg |
|---|---|---|---|---|---|---|---|---|
| DAN | Softmax | 76.03 | 67.05 | 67.28 | 54.58 | 75.39 | 80.05 | 64.15 |
| | Arcface | 66.14 | 61.51 | 57.63 | 51.01 | 66.34 | 72.44 | 56.02 |
| | Center | 75.74 | 66.99 | 67.17 | 54.60 | 75.21 | 79.71 | 63.90 |
| | UES | **77.43** | **71.81** | **70.67** | **58.92** | **78.25** | **83.14** | **68.20** |
| DANN | Softmax | 76.14 | 66.39 | 66.89 | 56.62 | 74.67 | 79.40 | 63.56 |
| | Arcface | 65.78 | 61.90 | 57.88 | 50.88 | 66.17 | 72.56 | 55.87 |
| | Center | 75.90 | 66.08 | 66.61 | 55.61 | 74.94 | 78.60 | 63.35 |
| | UES | **77.02** | **70.04** | **69.78** | **59.96** | **77.14** | **82.61** | **67.21** |
| DeepCORAL | Softmax | 76.43 | 66.29 | 67.70 | 53.56 | 75.39 | 79.55 | 63.71 |
| | Arcface | 66.41 | 62.07 | 57.74 | 50.54 | 66.61 | 72.44 | 55.80 |
| | Center | 76.16 | 66.39 | 67.72 | 53.70 | 75.29 | 79.47 | 63.55 |
| | UES | **77.28** | **70.95** | **70.63** | **57.99** | **78.11** | **82.59** | **67.65** |
| DAAN | Softmax | 76.10 | 65.95 | 67.50 | 53.01 | 74.69 | 79.34 | 63.05 |
| | Arcface | 66.03 | 62.10 | 57.75 | 49.88 | 66.71 | 72.73 | 55.79 |
| | Center | 75.80 | 65.99 | 67.07 | 52.54 | 74.76 | 78.83 | 62.93 |
| | UES | **77.33** | **70.93** | **70.50** | **57.25** | **77.84** | **82.56** | **67.42** |
| DSAN | Softmax | 76.15 | 69.59 | 69.81 | 59.17 | 76.31 | 81.36 | 66.17 |
| | Arcface | 71.74 | 66.54 | 61.64 | 53.88 | 72.13 | 76.13 | 60.61 |
| | Center | 76.24 | 69.88 | 69.54 | 59.38 | 76.56 | 81.05 | 66.22 |
| | UES | **77.69** | **71.81** | **71.19** | **59.56** | **78.44** | **83.21** | **68.78** |

Table 6: Accuracy (%) on office-Home for unsupervised domain adaptation

