# OpenReview forum: "UES: An Ultra-expanded Semantic Space for Unsupervised Domain Adaptation"
_ICLR.cc/2026/Conference — Submitted to ICLR 2026_

### Official Review · Reviewer_pixb · 2025-10-28

**Soundness:** 1
**Presentation:** 2
**Contribution:** 3
**Rating:** 2
**Confidence:** 4

**Summary:**

This paper proposes constructing an ultra-expanded semantic space for unsupervised domain adaptation (UDA) to improve both intraclass compactness and interclass separability. Unlike conventional margin-based methods that risk decision boundary violations and negative transfer, the approach pushes features away from decision boundaries to create a more generalizable metric space. It is easy to implement, compatible with existing feature alignment techniques, and achieves consistent performance improvements and enhanced testing stability.

**Strengths:**

1. It seems that the loss could be combined with any feature distribution alignment technique, showing its high generality and compatibility.
2. Simple design allows straightforward integration into existing models.
3. The paper is well-motivated and seems to be reproducible.

**Weaknesses:**

I believe this paper has two main limitations:
1. Although the authors introduce the proposed Ultra-expanded (UE) loss with good motivation, there seems to be a lack of in-depth theoretical discussion and analysis. As a result, readers may find it difficult to gain a truly insightful understanding. Please refer to [A].
2. The experiments are based on relatively outdated baselines. Improvements over these older methods may still fall short of the performance achieved by more recent approaches, even including zero-shot methods. It remains unclear whether the proposed methods would remain effective when applied to newer methods, especially those leveraging pretrained or large-scale models.

Other issues include:
1. The paper uses overly complex and non-standard notation, and contains several typos, which would benefit from careful proofreading.
2. The comparative experiments are somewhat limited, both in terms of baselines, datasets, and experimental settings.

[A] Xu, Gezheng, et al. "Revisiting Source-Free Domain Adaptation: a New Perspective via Uncertainty Control." The Thirteenth International Conference on Learning Representations. 2024.

**Questions:**

I hope the authors could provide further clarification or responses regarding the theoretical analysis and experimental settings.

---

> ### Author Response · Authors · 2025-11-28
>
> ·Q1:Although the authors introduce the proposed Ultra-expanded (UE) loss with good motivation, there seems to be a lack of in-depth theoretical discussion and analysis. As a result, readers may find it difficult to gain a truly insightful understanding.
>
>  A1: We thank the reviewer for recommending the relevant literature, which has enabled us to conduct a theoretical analysis and derivation of the UES loss from the perspective of generalization error bounds in Sections 3 and 4. Additionally, we have included a concise summary of generalization bounds in the Related Work section.
>
> ·Q2: The experiments are based on relatively outdated baselines. Improvements over these older methods may still fall short of the performance achieved by more recent approaches, even including zero-shot methods.
>
>  A2: We fully acknowledge this limitation in our experimental design. While we have incorporated an additional baseline method and expanded our evaluation to include the DomainNet dataset—with ongoing work addressing multi-source and source-free domain adaptation—we recognize that the current scope of our experiments remains insufficient.
>
> We have provided plug-and-play implementation code in Figure 4, and we kindly request if you could evaluate its performance within your project framework. Your findings—whether positive or negative—would be immensely valuable to us, as our current capacity for extensive experimental validation is limited. We would be honored to acknowledge your contribution in the acknowledgments section with sincere appreciation.
>
> ·Q3: The paper uses overly complex and non-standard notation, and contains several typos, which would benefit from careful proofreading.
>
>  A3: We have thoroughly rewritten the manuscript and revised all mathematical formulations. We sincerely apologize for any negative impact these issues may have had on your reading experience.
>
> ·Q4: I hope the authors could provide further clarification or responses regarding the theoretical analysis and experimental settings.
>
>  A4: We have explained these points in our responses to the official comments at the beginning of this review thread.

---

### Official Review · Reviewer_kizw · 2025-10-31

**Soundness:** 3
**Presentation:** 3
**Contribution:** 3
**Rating:** 4
**Confidence:** 3

**Summary:**

This paper presents a well-motivated and clearly written contribution to UDA. The idea of expanding classifier weights into an “ultra-expanded semantic space” is elegant and empirically validated. Although the method is simple, it provides notable and consistent gains over classical baselines with minimal complexity.

**Strengths:**

1.The proposed UES loss is conceptually simple yet effective. The theoretical motivation and geometric interpretation are clearly articulated, and the visualizations convincingly show its ability to enhance intra-class compactness and inter-class separability across domains.
2.The method introduces very few hyper-parameters, shows strong robustness to them, and can be easily integrated as a plug-and-play regularizer into existing UDA frameworks.
3.The Semantic Zoom mechanism is intuitively reasonable: using different temperatures for source and target domains is a neat way to retain semantically meaningful information while mitigating domain bias.

**Weaknesses:**

1.While the chosen alignment backbones (DAN, DANN, DeepCORAL, DAAN) are classic and representative, they are relatively old. The paper lacks comparison with more recent or stronger baselines, making it difficult to assess the broader applicability of UES in modern settings.
2.The margin-loss comparison focuses on Softmax, ArcFace, and Center Loss, which are also dated. The effectiveness of UES against more advanced discriminative losses remains unclear.
3.The Geometric Interpretation in Section 4.1 is insightful and explains how Wbasis enlarges the margin and thus enhances robustness. However, the analysis remains primarily at a geometric and empirical level, lacking a formal connection to existing domain generalization bounds or theoretical guarantees. Strengthening this link would substantially improve the paper’s rigor.

**Questions:**

1.The Semantic Zoom module employs different softmax temperatures. How sensitive is the approach to this setting, and would a learnable or adaptive temperature improve performance further?
2.Since UES constructs an expanded feature space, does it risk over-dispersion or feature instability in high-dimensional domains?
3.The experiments show that the method consistently improves accuracy and reduces A-distance, but it remains unclear how UES interacts with domain alignment losses (e.g., adversarial or moment matching). Does UES primarily benefit from better intra-domain discrimination, or does it also implicitly reduce inter-domain discrepancy?

---

> ### Author Response · Authors · 2025-11-28
>
> ·Q1: While the chosen alignment backbones (DAN, DANN, DeepCORAL, DAAN) are classic and representative, they are relatively old. The paper lacks comparison with more recent or stronger baselines, making it difficult to assess the broader applicability of UES in modern settings. The margin-loss comparison focuses on Softmax, ArcFace, and Center Loss, which are also dated. The effectiveness of UES against more advanced discriminative losses remains unclear.
>
>  A1: We acknowledge this limitation in our baseline selection. While we have added a 2020 method and DomainNet experiments, with multi-source adaptation underway, we recognize these remain insufficient. As visible in the public ICLR review comments, we hope to leverage our exceptional reviewers' expertise to strengthen this work. Any assistance would be gratefully acknowledged. Regarding margin losses, we've updated related work but found no theories offering better domain adaptation performance.
>
> ·Q2: The Geometric Interpretation in Section 4.1 is insightful and explains how Wbasis enlarges the margin and thus enhances robustness. However, the analysis remains primarily at a geometric and empirical level, lacking a formal connection to existing domain generalization bounds or theoretical guarantees. Strengthening this link would substantially improve the paper’s rigor.
>
>  A2: We thank the reviewer for their insightful comments. In Sections 3 and 4 of our paper, we have theoretically derived the UES method based on the margin-loss generalization bound. Additionally, we have included a concise summary of generalization bounds in the related work section.
>
> ·Q3: The Semantic Zoom module employs different softmax temperatures. How sensitive is the approach to this setting, and would a learnable or adaptive temperature improve performance further?
>
>  A3: Upon reading your comment, we were reminded of CLIP[A] and immediately implemented a learnable temperature parameter. Our experiments confirmed that this adaptive approach can indeed yield modest performance improvements. However, since the core thesis of our paper focuses on pushing features away from decision boundaries, we ultimately decided to remove the Semantic Zoom module as it represented an auxiliary engineering technique. We sincerely appreciate your suggestion, which has stimulated valuable further thinking on our part.
>
>
> ·Q4: Since UES constructs an expanded feature space, does it risk over-dispersion or feature instability in high-dimensional domains?
>
>  A4: Your question ultimately prompted us to reconsider the hyperparameter design, leading to a configuration that requires virtually no manual tuning.
>
> To investigate this concern, we inserted a 10,000-dimensional matrix before the linear classification layer, dramatically increasing the original feature dimension from 2,048 (ResNet) to 10,000. We conducted experiments on the OfficeHome Rw→Pr task using this high-dimensional setup. Initially, the performance dropped from 83.62 to 81.68 with UES applied, seemingly confirming the risk of over-dispersion.
>
> However, further reflection revealed that the expanded space constructed by UES can effectively occupy broader regions by scaling $\rho W$ to radiate larger areas in the high-dimensional space, thereby preventing feature dispersion. By adjusting $\rho$  from 300 to 700, performance improved from 81.68 to 82.63. Although this remains slightly below the original 83.62, the gap is attributable to overfitting, as indicated by the baseline performance drop from 81.36 to 80.94 without UES under the same high-dimensional condition.
>
> These findings led us to recognize the relationship between $\rho$ and the dimensionality of the feature space. As analyzed experimentally in Section 5.3, we established that ρ can be set dimension-dependent, eliminating the need for manual tuning. Therefore, UES requires only a single task-prior-based trade-off hyperparameter.
>
> ·Q5: The experiments show that the method consistently improves accuracy and reduces A-distance, but it remains unclear how UES interacts with domain alignment losses (e.g., adversarial or moment matching). Does UES primarily benefit from better intra-domain discrimination, or does it also implicitly reduce inter-domain discrepancy?
>
> A5: We thank the reviewer for this insightful comment. In Section 6, we provide a detailed analysis of these two issues from two distinct perspectives: the optimization behavior of the loss and further examination of intra/inter-class distances. 1.We observe that UES provides additional supervisory signals while simultaneously reducing the domain confusion loss. 2.UES primarily benefits from enhanced inter-class separability, which constitutes the fundamental rationale for applying our proposed margin-based method to UDA.
>
> [A]Radford, Alec,et al."Learning transferable visual models from natural language supervision." In International conference on machine learning. 2021.

---

### Official Review · Reviewer_kFiE · 2025-10-31

**Soundness:** 2
**Presentation:** 2
**Contribution:** 1
**Rating:** 4
**Confidence:** 4

**Summary:**

The paper proposes UES (Ultra-Expanded Semantic Space), a new loss formulation for Unsupervised Domain Adaptation (UDA). The authors claim that this loss can serve as a plug-and-play regularizer for existing UDA frameworks such as DANN, DAN, DeepCORAL, and DAAN. Experiments on three benchmark datasets (Digits, Office-31, Office-Home) show improved accuracy compared to ArcFace and Center Loss.

**Strengths:**

1. This paper attempts to design a simple, general-purpose regularizer applicable across UDA frameworks.
2. The figures in this paper are easy to understand.

**Weaknesses:**

1. All baselines (DANN, DAN, DeepCORAL, DAAN) are from 2015–2019. Missing modern methods makes the claimed superiority unconvincing.
2. The related work section is outdated and clearly misses recent advances in UDA from the past few years.
3. The experiments are limited to relatively simple datasets and lack evaluations on more challenging benchmarks such as DomainNet, which is commonly used in UDA research.
4. The writing quality requires further improvement. The paper should be reorganized for better readability, and the main experiments should be integrated into the main text rather than placed in the appendix.

**Questions:**

Please refer to Weaknesses.

---

> ### Author Response · Authors · 2025-11-28
>
> ·Q1: All baselines (DANN, DAN, DeepCORAL, DAAN) are from 2015–2019. Missing modern methods makes the claimed superiority unconvincing.
>
>  A1: We have incorporated a method from 2020 as an additional baseline and are currently preparing experiments on multi-source domain adaptation. We acknowledge that these additions may still fall short of your expectations, which we attribute solely to our own limitations in time and resources. We want to assure you that we are not evading any issues.
>
> The complete UES code is provided in Figure 4, which can be readily integrated into existing models. As an esteemed expert in this field, we would be profoundly grateful if you could evaluate our method within one of your projects and share the results with us—whether positive or negative. We would deeply appreciate your valuable feedback.
>
> ·Q2: The related work section is outdated and clearly misses recent advances in UDA from the past few years.
>
>  A2: We have updated the Related Work with a unique synthesis not found in prior surveys. We believe this version will meet your expectations and welcome further suggestions.
>
> ·Q3: The experiments are limited to relatively simple datasets and lack evaluations on more challenging benchmarks such as DomainNet, which is commonly used in UDA research.
>
>  A3: We thank the reviewer for this valuable suggestion. We have now supplemented our experimental evaluation on the DomainNet dataset, as detailed in Section 5.2.
>
>  Q4:The writing quality requires further improvement. The paper should be reorganized for better readability, and the main experiments should be integrated into the main text rather than placed in the appendix.
>
>  A4: We have undertaken a comprehensive revision of the manuscript to enhance its overall readability and structural organization. Following your recommendation, the DomainNet experimental results have been integrated into the main body of the paper, while results from the other three datasets are now presented through a clearer comparative visualization in Figure 4 (left). We sincerely appreciate your time and effort in reviewing our work.

---

### Official Review · Reviewer_zXJC · 2025-10-31

**Soundness:** 3
**Presentation:** 3
**Contribution:** 3
**Rating:** 4
**Confidence:** 5

**Summary:**

The paper proposes an innovative approach for Unsupervised Domain Adaptation (UDA) by introducing the Ultra-expanded Semantic (UES) loss, designed to enhance both intra-class compactness and inter-class separability. The approach aims to mitigate issues caused by the traditional FC + Softmax + Cross-Entropy loss, particularly in non-IID settings where features from the target domain exhibit large intra-class variation, leading to poor generalization. The UES loss incorporates an Ultra-expanded (UE) loss term and a Semantic Zoom mechanism, both of which are used to push features further from decision boundaries while preserving critical semantic information. Extensive experiments on several datasets demonstrate the proposed method's effectiveness, showing consistent improvements over baseline methods, such as ArcFace and Center Loss.

**Strengths:**

- Innovative approach: proposes a new margin-based UES loss that simultaneously enhances intra-class compactness and inter-class separability for UDA.

- Clear motivation: identifies the weakness of FC+Softmax+CE under non-IID assumptions and provides a reasonable explanation for why UES loss can help.

- Strong experimental validation: consistently outperforms ArcFace and Center Loss across multiple datasets and baselines (DAN, DANN, DeepCORAL, DAAN).

- Good robustness: demonstrates stable convergence and wide tolerance to hyperparameters (e.g., expansion factor e, λ₂).

- Easy integration: the loss is simple and can be added as a regularization term to existing frameworks with minimal code changes.

**Weaknesses:**

- Limited theoretical analysis: the paper lacks a solid theoretical justification for why UES loss performs better than other margin-based losses.

- Insufficient ablation studies: only limited analysis on the contribution of Semantic Zoom vs. UE loss; unclear which component drives most of the gains.

- Missing computational analysis: claims low overhead but provides no quantitative evidence (training time, memory, etc.).

- No discussion on limitations: does not explore scenarios where the method might fail (e.g., extreme domain shift or noisy target data).

- Writing and structure: some sections (especially “Inspirational Discoveries”) read more like extended discussion rather than rigorous analysis, which affects clarity.

**Questions:**

- Could the method be extended to multi-source domain adaptation? How does the UES loss behave when there are multiple source domains?

- How does the choice of hyperparameters (e.g., expansion factor, temperature) influence the results across different tasks? A more thorough analysis of hyperparameter sensitivity would be beneficial.

- Could the method be applied to more complex domains beyond image classification, such as natural language processing or video processing?

---

> ### Author Response · Authors · 2025-11-28
>
> ·Q1: Limited theoretical analysis: the paper lacks a solid theoretical justification for why UES loss performs better than other margin-based losses.
>
>  A1: Thank you for your insightful suggestion. In Section 6, we have conducted comprehensive analysis and experiments from two perspectives: the optimization behavior of the loss function, and the enhancement of intra-class compactness alongside inter-class separability.
>
> ·Q2: Insufficient ablation studies: only limited analysis on the contribution of Semantic Zoom vs. UE loss; unclear which component drives most of the gains.
>
>  A2: In the revised manuscript, we have derived the Ultra-Expanded Space (UES) loss from the margin-based generalization bound. The Semantic Zoom mechanism has been removed as it constituted an engineering technique built upon the UES loss. This streamlines our approach, focusing it squarely on the core contribution of pushing features away from decision boundaries. We confidently affirm that the Ultra-Expanded Space loss represents the primary contribution of our proposed method.
>
> ·Q3: Missing computational analysis: claims low overhead but provides no quantitative evidence (training time, memory, etc.).
>
>  A3: We have addressed the computational analysis concerns through the following evidence:
> First, we provide the source code in Figure 4 (right), which demonstrates that UES introduces only one additional tensor squared-distance operation. Subsequently, we present per-epoch training time and convergence analysis in Figure 5 (left). Remarkably, UES not only maintains convergence properties but unexpectedly reduces training time. This improvement can be attributed to the UES loss potentially enabling PyTorch's computational graph optimizer to identify and exploit more operator fusion opportunities, thereby enhancing overall execution efficiency. Please refer to Section 5.3 "Convergence and Runtime" .
>
> ·Q4: No discussion on limitations: does not explore scenarios where the method might fail (e.g., extreme domain shift or noisy target data).
>
>  A4: Your thought process mirrors my initial confusion. When the experimental results came in, I was initially surprised by how extremely positive they were. The previous results were on Digit, Office31, and OfficeHome. We supplemented the difficult dataset Domainnet and found that only one item showed a slight decrease. We analyzed the reasons in section 5.2.
>
> ·Q5: Writing and structure: some sections (especially “Inspirational Discoveries”) read more like extended discussion rather than rigorous analysis, which affects clarity.
>
>  A5: We thank the reviewer zXJC for this constructive feedback. We have thoroughly revised the manuscript to enhance its analytical rigor and structural clarity. Should any sections remain unclear, we would be grateful for specific guidance and are prepared to undertake further revisions.
>
> ·Q6: Could the method be extended to multi-source domain adaptation? How does the UES loss behave when there are multiple source domains? Could the method be applied to more complex domains beyond image classification, such as natural language processing or video processing?
>
>  A6：We thank the reviewer for these insightful questions regarding the method's broader applicability. We are currently preparing supplementary experiments on multi-source domain adaptation. In Section 4.3, we provide gradient analysis demonstrating that the UES loss offers more stable optimization compared to entropy minimization. Entropy minimization is widely adopted in UDA to enhance prediction confidence and promote cross-domain distribution alignment. Our proposed UES loss provides a theoretically superior alternative to this established approach. Empirical validation on image classification tasks is provided in Section 5.3 (UES vs. Entropy Minimization).
>
> We acknowledge that without final experimental results across diverse domains, these claims may require further substantiation. Due to current constraints in time and resources for reproducing additional tasks, we would like to respectfully request your assistance. Given your expertise in the field, we have made our complete source code available in Figure 4. We would be grateful if you could evaluate its performance on more complex domains.We would be pleased to acknowledge your contribution in the acknowledgments section for any insights you might share.
>
> ·Q7: How does the choice of hyperparameters (e.g., expansion factor, temperature) influence the results across different tasks? A more thorough analysis of hyperparameter sensitivity would be beneficial.
>
>  A7: Your analysis is highly insightful. Building upon Reviewer kizw's comments, we have gained new inspiration and conducted a detailed analysis in Section 5.3 (Hyperparameter). A significant improvement in this revision is that our method now requires virtually no hyperparameter tuning.

---

### Author Response · Authors · 2025-11-28

We thank all reviewers for their valuable comments and constructive feedback. We have diligently addressed the concerns raised and significantly improved the manuscript accordingly. Although the score adjustment system has now closed, your insightful comments have substantially enhanced the quality of our manuscript, which truly embodies the fundamental purpose of academic research. We would be grateful if you could review our revised manuscript. Below we provide a point-by-point response to the two major issues highlighted:
### Theoretical Foundation:
Following the reviewers’suggestions, we have supplemented the theoretical derivation based on the margin generalization bound. We want to create intuitive articles like Resnet, which were later theoretically proven effective and proposed further improvements. Erwin Schrödinger derived the Schrödinger equation, and Max Born's explanation of the Schrödinger equation using the concept of probability waves gave it unparalleled significance and earned it the Nobel Prize in Physics. Thank you to reviewer zXJC for proposing a comparison with other margin loss based methods, Kizw for proposing a formal connection with the generalization bound or theoretical guarantee, and Pixb for providing the article. We have extensively considered the generalization error bound and described a portion of it in related work.
### Experimental Evaluation:
We have expanded our experiments by including results on the DomainNet dataset suggested by Reviewer kFiE and the new baselines  Additionally, we are currently preparing multi-source domain adaptation experiments. If there is still time, we will supplement semantic segmentation.Due to time constraints, We recognize that the current experimental scope remains insufficient to fully satisfy the reviewers' intellectual curiosity regarding our method's capabilities. Therefore, **we have added the plug and play PyTorch code (“Talk this cheap,show me the code”)proposed in this paper in Figure 4, which shows that our proposed method is very concise.** Theoretically, our approach is applicable to any task involving linear-layer classification, including semantic segmentation (pixel-level), object detection (region-level), and language modeling (token-level). **We sincerely hope reviewers will consider integrating our method Ultra-Expanded Space
(UES) into their UDA projects and welcome any feedback on its empirical performance. We appreciate the ICLR Alliance for providing us with the opportunity to exchange ideas and showcase the outstanding abilities of reviewers. We will cite this website in our acknowledgements and sincerely thank all the reviewers.**

Recognizing the considerable demands on the reviewers' time and attention during this period, We will provide concise and targeted responses to each comment to facilitate an efficient evaluation process.With the rebuttal deadline approaching, we will remain fully available to address any further questions or concerns the reviewers may have.

We extend our best regards to the review committee.

---

> ### Author Response · Authors · 2025-11-29
>
> In Section 5.2, we have supplemented our evaluation with multi-source domain adaptation experiments, which demonstrate consistent performance improvements. Notably, these results were achieved using the **exact hyperparameters from Figure 4 without any modifications**.

---

> > ### Author Response · Authors · 2025-12-04
> >
> > In Section 5.2, we additionally conducted a semantic segmentation experiment without modifying any hyperparameters, which similarly led to improved results. We strongly recommend that researchers integrate the code corresponding to Figure 4 of this paper into their own projects, as it has proven to be highly effective and practical.

---

### Meta-Review · Area_Chair_WfrT · 2026-01-08

**Summary:**

Reviewers felt the theoretical justification was still not as rigorous as modern margin-based and generalization-bound analyses. They found key baselines remained outdated, limiting evidence of competitiveness in current UDA settings. The experiments were still seen as incomplete, with missing ablations and limited validation despite the added results.

**Reviewer Concerns:**

Addressed:
* Clarity and writing improvements
* Computational overhead clarification
* Additional experiments for runtime and convergence analysis and ablation study, on new datasets.

Outstanding:
* Whether the baseline methods are competitive
* Whether the validation is complete and mature
* Whether the theoretical justification is strong and grounded.

**Reviewer Scores:**

pixb provided the lowest rating initially, and might not change the rating even after the full discussions. All other reviewers rated this paper as "marginally below accept" and may raise their ratings by 1-2 since the response addressed (at least partially addressed) their concerns, such as weak theoretical contributions and experiments.

---

### Decision · Program_Chairs · 2026-01-26

Reject